# Molecular Characterisation of Epstein–Barr Virus in Classical Hodgkin Lymphoma

**DOI:** 10.3390/ijms232415635

**Published:** 2022-12-09

**Authors:** Valerija Begić, Petra Korać, Slavko Gašparov, Marija Rozman, Petra Simicic, Snjezana Zidovec-Lepej

**Affiliations:** 1Division of Molecular Biology, Department of Biology, Faculty of Science, University of Zagreb, 10000 Zagreb, Croatia; 2Primary School “Sesvetski Kraljevec”, 10361 Sesvetski Kraljevec, Croatia; 3Institute of Clinical Pathology and Cytology, Merkur University Hospital, 10000 Zagreb, Croatia; 4Department of Pathology, Medical School Zagreb, University of Zagreb, 10000 Zagreb, Croatia; 5Department of Immunological and Molecular Diagnostics, University Hospital for Infectious Diseases, 10000 Zagreb, Croatia

**Keywords:** EBV, Hodgkin’s lymphoma, LMP1, genotyping

## Abstract

Hodgkin lymphomas (HLs) are a heterogeneous group of lymphoid neoplasia associated with Epstein–Barr virus (EBV) infection. EBV, considered to be an important etiological co-factor in approximately 1% of human malignancies, can be classified into two genotypes based on EBNA-2, EBNA-3A and EBNA-3C sequences, and into genetic variants based on the sequence variation of the gene coding for the LMP1 protein. Here, we present the results on the distribution of EBV genotypes 1 and 2 as well as *LMP1* gene variants in 50 patients with EBV-positive classical HL selected from a cohort of 289 histologically verified cases collected over a 9-year period in a tertiary clinical center in the Southeast of Europe. The population-based sequencing of the *EBNA-3C* gene showed the exclusive presence of EBV genotype 1 in all cHL samples. The analysis of EBV LMP1 variant distribution showed a predominance of the wild-type strain B95-8 and the Mediterranean subtype with 30 bp deletion. These findings could contribute to the understanding of EBV immunobiology in cHL as well as to the development of a prophylactic and therapeutic vaccine.

## 1. Introduction

Hodgkin lymphomas (HLs) are a heterogeneous group of lymphoid neoplasia that are derived from B-cells and histologically consist of large neoplastic cells (CD30^+^ Hodgkin and Reed–Sternberg cells, HRS) and a diverse group of reactive bystander cells (histiocytes, lymphocytes, plasma cells, epitheliod histiocytes, epithelioid granulomas and eosinophils, depending on the HL type [1]. Based on a WHO classification system [2,3], HL includes two major pathological types: classical HL (cHL), which represents about 90% of cases, and a nodular lymphocyte-predominant HL (NLPHL), which exhibits partially overlapping biological features with T-cell/histiocyte-rich large B-cell lymphoma (THRLBCL) [4]. Based on histological and molecular features, cHL is classified into four types: lymphocyte-rich (LR), lymphocyte-depleted (LD), mixed-cellularity (MC) and nodular sclerosis (NS) [1]. The most frequent cHL type is NSCHL (70–80% of cases in developed countries) followed by MCCHL (20–25%), LRCHL (5%) and the rarest type, LDCHL (<2%) [4].

Epstein–Barr virus (EBV) or *Human gammaherpesvirus 4* is a member of the *Herpesviridae* family (*genus Lymphocryptovirus*) that is considered a prototypic oncogenic virus enabling the transformation of B-cells into B-lymphoblastoid cell lines in vitro by using complex molecular mechanisms mediated by oncogenic proteins and microRNAs (miRNAs) [5]. The EBV genome consists of a linear double-stranded DNA of approximately 170 kb that includes 80 possible coding regions, a series of repeat regions (terminal direct repeats and internal repeat sequences) as well as large indel regions [5]. EBV infects more than 90% of the human population but causes illness in a relatively limited percentage of persons. Diseases associated with EBV infection cause significant morbidity and mortality and the virus is considered to be an important etiological co-factor in approximately 1% of human malignancies [6]. However, prophylactic or therapeutic EBV vaccines are currently not available [7].

The frequency of EBV detection in different cHL types depends on age and geographical distribution. In developed countries, the EBV-positivity rate is the highest for the MCCHL and LDCHL (estimated at 75%), but substantially lower for LRCHL (30–50%) and NSCHL (10–25%) (for a review see Satou et al., 2020). According to the recent estimate on the burden of EBV-attributed malignancies using data from the Global Burden of Disease Study, there were 101,133 cases of EBV-associated HLs and 32,560 deaths in 2017 [8]. The immunobiology of EBV in the context of the neoplastic transformation of cells in cHLs has been extensively studied. HRS cells that originate from B-lymphocytes within the germinal centers of lymph nodes are universally infected with EBV [4]. Infected HRS cells exhibit a latency II pattern of the EBV replication cycle that is characterized by the expression of latent membrane proteins (LMP)-1 and -2A, EBV nuclear antigen (EBNA) 1 and EBV-encoded small RNAs. LMP1 and LMP-2A play an important role in the transformation of B-cells.

LMP1 is considered to be an essential oncogenic protein of EBV since a recombinant virus lacking this protein is unable to immortalize resting B-cells [9]. It is a 356-amino acid integral membrane protein that mimics a physiological signal transmitted by a molecule CD40 that leads to the activation of the NF-κB, PI3K/AKT and JAK/STAT signal transduction pathways. The constitutive activation of these signaling pathways by LMP1 subsequently rescues B-cells from apoptosis and enables their survival and proliferation [9,10].

EBV is classified into two genotypes (based on sequences of genes coding for latent proteins EBNA-2, EBNA-3A and EBNA-3C) that exhibit different geographical distributions and distinct immunobiological features in vitro and in animal models [11,12]. Furthermore, EBV is classified into genetic variants based on the sequence variation of the gene coding for LMP1 (C-terminal part), relative to the wildtype strain B95-8. LMP1 variants of EBV include Alaskan (AL), China1, China2, China3, Mediterranean with (Med+) or without (Med-) deletions, and North Carolina (NC) with some infrequently described variants as well (SEA1, SEA2 and CAO) [13,14]. In addition, an EBV molecular classification based on 12 EBV phylopopulations of monophyletic and paraphyletic origins that was obtained by using a Bayesian analysis of the population structure based on the alignments of masked genomes has also been described [15]. Multiple specific LMP1 variants have been described in benign and malignant EBV-associated diseases have been described but the data on their clinical relevance are still unclear [11,12,16,17,18,19,20,21,22,23,24,25,26,27,28,29,30,31,32,33,34,35,36,37,38,39,40,41,42,43]. The aim of this study was to analyze the distribution of EBV genotypes 1 and 2 as well as EBV LMP1 variant diversity in patients with cHL from the Southeast of Europe in the context of clinical and pathological findings.

## 2. Results

### 2.1. Patients and Methods

Out of 289 cHL samples, 50 were EBV-positive. EBV infection was detected based on the analyses of LMP1 protein (using immunohistochemical staining) or EBER (using ISH). (Figure 1). Out of 50 EBV-positive samples, 46 EBNA3C and LMP1 sequences gained by Sanger sequencing had adequate quality for further analysis. Among those 46 patients, there were 34 men and 12 women. The age range was between 10 and 87 years with a median of 56. There were 19 cases of MC, 11 cases of NS, 3 cases of LR, 1 case of LD, 3 SLL transformations to cHL, and 9 cases for which the subtype could not be determined (Table 1). Some of the clinical data were available for 30 patients. According to the Ann Arbor staging system, 21 patients belonged to a lower stage (I-III), while 8 belonged to a higher stage (IV). Most patients (19) were treated with ABVD (doxorubicin, bleomycin, vinblastine, dacarbazine), and all patients (23) who received therapy and for whom data were available achieved full remission. Follow-up was carried out between 0 and 99 months (Appendix A).

### 2.2. EBV Variants and Clinical Data

Out of 50 EBV-positive samples, 46 EBNA3C sequences gained by Sanger sequencing had adequate quality for further analysis. All samples were determined as the EBV-1 genotype according to the EBNA3C sequence. Out of 50 EBV-positive samples, 46 LMP1 sequences gained by Sanger sequencing had adequate quality for further analysis. Most cases belonged to the wild-type (24) and Mediterranean subtype with 30 bp deletion (13). Mediterranean subtype without 30 bp deletion, North Carolina subtype, and coinfection followed (3, 2 and 3, respectively). Out of three coinfections, one was a wild-type/Mediterranean subtype without 30 bp deletion, and two were wild-type/North Carolina subtype. Additional cases could not be specified according to the algorithm by Edwards [13] (Table 2). There were no statistically significant differences between age, sex, cHL subtypes and EBV subtypes (*p* > 0.05). The survival rates did not differ among specific EBV subtypes (*p* > 0.05).

## 3. Discussion

The molecular diversity of EBV in various malignant and non-malignant diseases is exceptionally important for a more complete understanding of viral immunobiology, the development of innovative therapeutic strategies as well as for innovative approaches to prophylactic and therapeutic vaccine development. In this study, we present results on the distribution of EBV genotypes 1 and 2 as well as *LMP1* gene variants in 46 patients with EBV-positive cHL selected from a cohort of 289 histologically verified cases collected over a 9-year period in a tertiary clinical center in the Southeast of Europe. The population-based sequencing of the gene coding for EBNA-3C showed the exclusive presence of EBV genotype 1 in all bioptic samples from cHL patients. The analysis of EBV LMP1 variant distribution showed a very narrow repertoire with a predominance of the wild-type strain B95-8 and Mediterranean subtype with 30 bp deletion. Other EBV LMP1 variants, including the North Carolina and Mediterranean subtypes without 30 bp deletion as well as co-infections with the wild-type B95-8, were exceptionally rare. The EBV LMP1 variants Alaskan, China1, China2 and China3 were not detected in our cohort. In vitro studies on the biological differences between EBV type 1 (e.g., B95-8) and type 2 (AG876 strain) have shown the superior capacity of EBV type 1 to transform B-cells into lymphoblastoid cell lines (LCLs) compared to type 2 [44]. However, in vivo studies in a BALB/c Rag2null IL2rnull SIRP humanized mice model have shown the ability of EBV type 2 to induce a tumor similar to diffuse large B-cell lymphoma and establish a type III latency in B- and T-cells, suggesting that EBV type 2 exhibits oncogenic properties in vivo [45]. Simbiri et al. (2015) detected the type 2 genotype in a lymphoblastoid cell line generated from a sample obtained from a child with Burkitt’s lymphoma from sub-Saharan Africa [46]. These studies confirm the ability of EBV type 2 to induce oncogenesis in vivo. EBV genotypes also exhibit differences in viral tropism with recent experiments showing that genotype 2 has the ability to infect mature human CD3+ T-cells in vitro and can be detected in purified T-cells in the pediatric populations of Kenya ex vivo [47,48].

The analysis of EBV type 1 and 2 distribution in various populations (endemic Burkitt’s lymphoma, immunocompetent healthy persons from diverse age groups with various sexual practices, infectious mononucleosis patients, transplanted patients and patients with malignant diseases) across different geographic areas reveals a complex pattern with a global predominance of type 1 infection (for review see Appendix A) [23,25,26,29,32,33,38]. The prevalence of EBV types 1 and 2 in HL has been investigated in several studies from various parts of the world (Appendix A). Wang et al. (2021) recently showed a clear predominance of type 2 EBV in 98.6% of patients with non-Hodgkin’s lymphoma (n = 46) and Hodgkin’s lymphoma (n = 16) from China with one mixed type 1/type 2 infection [40]. Similarly, the analysis of lymph node biopsy specimens in 42 children and 16 adults with HL from Mexico showed a high frequency of type 2 infection (47.6% in children and 69.2% in adults) with mixed infections present in 19% of children [30]. Contrary to these findings, the majority of studies from other parts of the world showed a predominance of EBV type 1 in HL. Salahuddin et al. (2018) showed the predominance of EBV type 1 in tissue samples in patients with various types of lymphoma from Pakistan (90.7%), similar to the results by Kafita et al. (2018) obtained by analyzing biopsy samples from 150 cases of lymphoma (80% NHL, 20% HL) [34,35]. In Iran, Tabibzadeh et al. (2021) showed a predominance of type 1 infections (91.2%) in the peripheral blood of 34 patients with various types of malignant diseases including 8 patients with HL [37] (Appendix A). Our results have shown an exclusive presence of EBV type 1 in biopsy samples from cHL patients from Croatia. The heterogeneity of the data in the literature on this issue confirms a need for a more extensive analysis of EBV genotype 1 and 2 distribution in various geographic regions with a particular emphasis on patients with different types of malignant diseases.

The distribution of EBV LMP1 variants has been investigated in various populations including healthy young adults, patients with IM as well as in patients with various malignant diseases including HL, non-HL and NPC (Appendix A) [16,17,18,19,21,24,29,33,42,43]. The molecular diversity of EBV LMP1 in HL has been analyzed in five studies published in the period 1993–2012 that focused on heterogeneous patient groups and samples reporting various patterns of variants (Xhol loss, 30 bp del, Mediterranean without deletion) (Appendix A) [16,18,19,21,24,29]. The largest of these studies, by Knecht et al. (1993), analyzed LMP1 gene diversity in a cohort of 52 HL patients mainly classified MC (n = 25 cases) and NS types (n = 23). EBV LMP1 wild-type was present in almost all samples (90.4%) with deletions clustered within the 75-bp region at the 3′ end of the *LMP1* gene in five cases belonging to the MC type exhibiting numerous or abundant HRS cells. Contrary to these findings, our results showed that about half of cHL cases were associated with the wild-type infection with a much higher proportion of Mediterranean subtype with 30 bp deletion as well as the presence of the North Carolina and Mediterranean subtypes without deletion showing a much higher molecular diversity of EBV in cHL than previously thought. Recently, Alves et al. (2022) showed a high degree of LMP variant diversity in a study on 33 patients with malignant diseases diagnosed between 1995 and 2007 including biopsy specimens from 26 cHL patients, 7 BL patients and 4 patients with reactive hyperplasia from Brasil as well as 42 healthy persons with detectable EBV DNA in their saliva [42]. The study showed the predominance of the Mediterranean (40.2%, n = 33) and Raji/Argentine (39%, n = 32) variants while the prototype virus B95-8 was detected in only five samples and the Asian II variant in one sample. The remaining sequences (13.4%) did not group into well-defined clades with other reference strains. Of note, the Raji/Argentine clade carrying polymorphisms I124V/I152L, del30 bp and ins15 bp was present in 61% of lymphomas. These results highlight a specific pattern of LMP1 variants with a clear predominance of Raji/Argentine LMP1 characteristics for patients with lymphomas from Brasil.

The distribution of EBV LMP1 variants in other types of malignancies also showed characteristic patterns. A recent review by Montes-Mojarro et al. (2021) analyzed EBV LMP1 variant distribution in 6 studies (2 from Mexico and China, as well as studies from Peru and Argentina) that included a total of 140 patients with extranodal natural killer T-cell lymphoma (ENKTCL) which showed a clear predominance of the wild type (52.1%) with 37.1% of patients harboring 30 bp deletion [20,21,22,31,41,43]. These results show a high degree of similarity to our results in patients with cHL from Croatia. Sandvej et al. (1994) reported the exclusive presence of the 30 pb del variant in 9 cases of peripheral T-cell lymphoma (PTCL) from Malaysia as well as a higher proportion of 30 bp del mutant compared to the wild-type virus (61.1% vs. 38.9%) in a study on 18 patients from Denmark [17] (Appendix A). EBV LMP1 variants present in healthy persons and IM patients showed a much higher degree of diversity compared with patterns observed in EBV-associated malignancies as shown by Correia et al. (2017) and Banko et al. (2016) (Appendix A).

Mutations in the C carboxy-terminal region of LMP1 are associated with important changes in the transforming ability and immunogenicity of EBV. EBV strains carrying a 30 bp deletion near the 3′ derived from NPC (NCP1510 and NPC CAO) render LMP oncogene more biologically aggressive [49]. The transfection of non-tumorigenic human keratinocyte lines (Rhek-1) immortalized with a hybrid adenovirus 12-SV40 virus with the C-LMP variant that originated from nude mouse passaged NPC CAO showed a higher ability to grow in low serum concentrations and higher clonability with increasing LMP1 expression as well as oncogenic potential in vivo demonstrated by the ability to induce invasive tumors in SCID mice [49]. The increased oncogenic potential of LMP1 with 30 bp deletion is not restricted to NPC as shown in studies on the association between the increased abundancy of HRS cells in HD, suggesting the increased ability of mutated LMP to induce their proliferation and increased histopathological severity in HD patients infected with a variant [16,50]. The increased histopathological potential of the LMP1 variants has been associated with the prolongation of the LMP1 half-life by 30–50% (demonstrated in mutants carrying 55 amino acid deletions in the C-terminal part). LMP1 amino acid residues 322 and 333 are required for the rapid turnover of the protein. Knecht et al. (1992) described the increased cytopathic effect of 30 bp mutants terminating at residues 322 and 333 that were probably associated with the reduced turnover of the protein, stronger LMP1 expression in HRS (including gigantic ones) and increased necrosis [16]. The loss of a restriction site in the exon 1 of the LMP1 gene, e.g., XhoI polymorphism frequently observed in NPC patients of Asian origin, has been associated with an increased oncogenic potential both in vitro (leading to the transformation of Balb/c 3T3 cells) and in vivo (a tumorigenic effect in nude mice) [51].

Mutations in the LMP1 gene are also associated with important differences in the immunogenicity of EBV which has important clinical implications regarding immune surveillance in oncology. LMP1 is expressed in the HRS of approximately 50% of HD and in malignant cells of about 60% of NPCs. Studies in LMP1-positive and -negative NPCs demonstrated increased immunogenicity of the B-cell-derived LMP1 isolate (wild-type B95-8) that exhibited a low mutagenic frequency in comparison with the highly mutated NPC CAO strain protein suggesting that LMP1-positive tumors escape immune surveillance by selecting for LMP1 mutations or similar mechanisms (the down-regulation of LMP1 expression, promotor methylation) allowing for immune evasion in immunocompetent hosts [52].

The prognostic role of LMP1 expression in cHL was evaluated in several studies. The expression of LMP1 as a marker of EBV infection in HL is associated with the mixed-cellularity subtype of cHL and a high-risk international prognostic score [53]. According to Hu et al. (2022), and their systematic review and meta-analysis of 42 qualified studies involving 9570 HL patients, there was an association between EBV positivity and significantly poorer overall survival, particularly in older patients [54]. Additionally, in a recent study by Nohtani et al. (2022), the expression of LMP1 in children with cHL was associated with a significantly lower risk of treatment failure in a Cox regression model [55]. Although there was no difference in overall survival between EBV-positive and -negative groups, EBV-positive children with Chl had significantly longer event-free survival [55]. The data in the literature on the impact of LMP1 variants including 30 bp mutants on overall survival, risk of treatment failure or time of event-free survival are currently not available.

The analysis of LMP1 molecular diversity in various malignant and non-malignant diseases associated with EBV is also important for the development of LMP1-targeted therapies. Therapeutic strategies in this field include: (1) LMP1-specific cytotoxic T-cells as a part of the adoptive cell transfer immunotherapy of EBV latency II-associated malignancies, (2) antibody-based immunotherapy using LMP1 peptide-specific immunization in LMP1-expressing tumors, (3) immune checkpoint inhibitor therapy (blocking the biological function of an LMP1 target PD-L1 with monoclonal antibodies in recurrent or metastatic NPC), (4) LMP1 sequence-specific RNA-cleaving DNAzime (such as DZ509) and (5) the development of small-molecule inhibitors targeting LMP1 (such as B1.12, ACPLDLRSPCG peptide or nanoparticle-conjugated EBNA1-LMP1-binding peptide) [56,57,58]. Of note, the development of therapeutic vaccines involving either dendritic cell or viral vector-based strategies (or both) so far mainly targeted EBNA1 and LMP2 proteins (not LMP1) due to their immunogenicity [59]. The results of our study have shown the presence of the Mediterranean subtype with 30 pb deletion in approximately 28% of patients with cHL, suggesting the need to evaluate the efficacy of the novel and/or developing LMP1-targeted therapies against this high-risk LMP1 variant both during drug development stages and in clinical trials. This study is an initial attempt to try to characterize EBV diversity in patients with malignant diseases (focusing on cHL) from Croatia that also exhibits limitations that need to be pointed out. One of the most important limitations in the LMP1 gene variability emphasized in many studies published so far is the heterogeneity of the analyzed populations including their geographic, ethnic, social and racial backgrounds [42].

This study presents data from Croatia, a small southeastern European country with a population estimated at 3,871,883 inhabitants (according to the 2021 census) and a Gross Domestic Product per capita of 17,398.8 USD [60,61]. Croatia is a country with a relatively homogenous ethnic structure consisting of Croats (91.6%) and Serbs (3.2%) with only 5.2% of persons of other non-Caucasian backgrounds and no significant immigration. Since human migration patterns have an important influence on the patterns of viral diversity in different parts of the world, the narrow patterns of EBV types and LMP1 variants observed in our study could reflect the abovementioned homogeneity of the local populations. Therefore, additional data on EBV diversity in cHL patient populations from different regions, ethnicities, and socioeconomic statuses are needed to obtain a more complete assessment of this issue and to evaluate an in vivo oncogenic potential of individual variants more objectively. In conclusion, we demonstrated an exclusive presence of EBV type 1 in the largest cohort of patients with histologically verified cHL from Europe with a narrow repertoire of LMP1 variants that includes the B95-8 wild type and Mediterranean subtype with 30 bp deletion. Despite the complex host, viral and environmental determinants involved in the pathogenesis of malignant diseases, the analysis of EBV diversity in various populations and geographic regions represents an important contribution to the understanding of EBV immunobiology.

## 4. Materials and Methods

### 4.1. Patients

This study included 289 cHL tissue samples from patients diagnosed at University Hospital Merkur between 2013 and 2021. All samples were formalin-fixed paraffin-embedded tumor tissue blocks taken from the hospital’s archive. The study was approved by the Ethics committee of the University Hospital Merkur, Zagreb, Croatia, on 25 September 2019.

### 4.2. Methods

Detection of EBV LMP1 protein or EBER (Epstein–Barr encoding RNA) in tumor tissue samples

EBV detection in tumor tissue samples was performed by immunohistochemical staining of LMP1 protein and/or in situ hybridization of EBER. In brief, immunohistochemical staining was performed on 2 µm thick tumor tissue sections. After the heat-induced epitope retrieval, incubation with primary anti-LMP1 antibody (Agilent Dako, Santa Clara, CA, USA) followed, and polymer-based detection system EnVision (Dako/Agilent, Santa Clara, CA, USA) was used according to the manufacturer’s instructions. Chromogenic detection was performed by secondary antibody conjugated with horseradish peroxidase after the addition of 3,30-diaminobenzidine.In situ hybridization was performed on 3 µm thick tumor tissue sections. Slides were deparaffinized in xylene substitution, rehydrated in decreasing concentrations of ethanol and incubated in proteinase K solution. After washing and dehydration in absolute ethanol, slides were dried, and EBER PNA probe (Dako/Agilent (Santa Clara, CA, USA) was added. Slides were incubated at 55 °C for 90 min, washed and anti-FITC/AP was added to samples that were then incubated for 30 min in humid conditions at room temperature. After washings, the substrate 5-bromo-4-chloro-3-indolylphosphate and nitroblue tetrazolium were added and slides were incubated for another 30 min. Samples were then stained with Nuclear Fast red dye (Dako/Agilent (Santa Clara, CA, USA).

Determination of EBV types 1/2 and LMP1 variant determination

DNA was isolated from two 10 µm thick FFPE sections of each case using commercially available kit Quick-DNA/RNA FFPE Miniprep Kit (ZymoResearch, CA, USA) according to manufacturer’s instructions. Fragments of *EBNA3C* and *LMP1* genes were amplified using previously described primers (Appendix A) [28,36]. Amplicons were purified using ChargeSwitch PCR Clean-Up Kit (Life technologies, CA, USA) and then sent to Macrogen (Republic of Korea) for Sanger sequencing. The Benchling program was used to overlap the forward and reverse sequences that were further aligned and analyzed using ClustalX2 and MEGA11 software. EBV genotype was determined using *EBNA3C* gene amplicon. Amplicons that were 153 bp long were classified as EBV-1 and amplicons that were 246 bp long were classified as EBV-2 according to Sample J et al. [62]. According to Edwards et al., LMP1 variants were identified using algorithm based on DNA sequence coding for a specific amino acid of the LMP1 C-terminus region [13] (Table 3).

### 4.3. Statistics

Chi-square test was used to determine the association between the sex, cHL subtypes and EBV subtypes, and Mann–Whitney U-test was used for the comparison of differences between age and EBV subtypes. OS (overall survival) was evaluated from the date of diagnosis to the date of death of any cause by the Kaplan–Meier method, and the comparison of OS between EBV subtypes was made by log rank test. Statistical analysis was performed with the STATISTICA software, version 13.0 (StatSoft Inc., Tulsa, OK, USA). Significance level was set at *p* < 0.05.

## Figures and Tables

**Figure 1 ijms-23-15635-f001:**
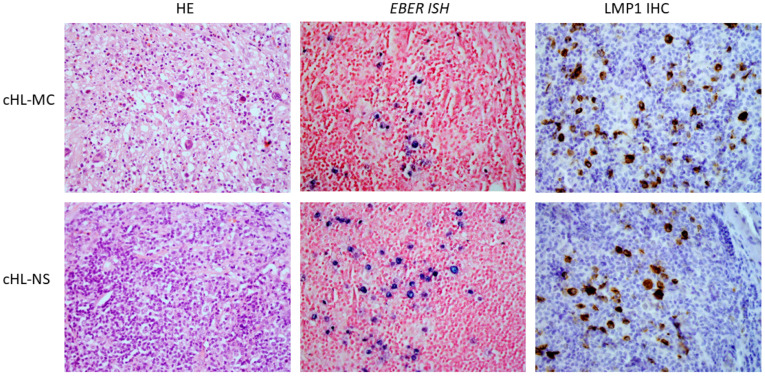
Most EBV-positive cases were mixed-cellularity (MC) and nodular sclerosis (NS) cHL subtypes. (HE-hemalum eosin; *EBER ISH*- in situ hybridization for detection of *EBER*, positive cells are stained blue, LMP1 IHC-immunohistochemical staining for detection of LMP1, positive cells are stained brown). Microscope’s magnification: 400×.

**Table 1 ijms-23-15635-t001:** EBV-positive patients with classical Hodgkin lymphoma (cHL).

Patients	Number of Samples	Age (Years)	Histological Subtype					
			MC	NS	LR	LD	SLL Transformations to cHL	Subtype Not Determined
Female	12	39–84 (median 67)	6	3	0	0	2	1
Male	34	10–87 (median 55)	13	8	3	1	1	8

Lymphocyte-rich (LR), lymphocyte-depleted (LD), mixed-cellularity (MC) and nodular sclerosis (NS) cHLs.

**Table 2 ijms-23-15635-t002:** EBV subtypes determined by *LMP1* sequence.

Sample	Histological Subtype	LMP1													EBV Genotype (EBNA3C)
		331 Gly-Gln/Ala	334 Gln-Arg	338 Leu-Ser/Pro	343–352 del	344 Gly-Asp	345 Gly-Ser	352 His-Arg/del/Asn	354 Gly-Asp	355 Gly-Ala/Thr/Val	358 His-Pro	366 Ser-Thr/Asn/Ala	del 30 bp	Subtype	
cHL1	NS	Gly	Pro	Leu	WD	Gly	Gly	His	Gly	Gly	His	Thr	WD	B95-8	EBV-1
cHL2	MC	Gly	Pro	Leu	WD	Gly	Gly	His	Gly	Gly	His	Thr	WD	B95-8	EBV-1
cHL3	MC	Gly	Gln	Leu	WD	Gly	Gly	His	Gly	Gly	His	Thr	WD	B95-8	EBV-1
cHL5	MC	Gly	Arg	Ser	WD	Gly	Gly	Arg	Gly	Gly	His	Thr	WD	Med−	EBV-1
cHL6	MC	Gly	Arg	Leu	/	Gly	Gly	del	/	/	His	Thr	del	Med+	EBV-1
cHL7	NS	Gly	Arg	Ser	/	/	/	del	Gly	Gly	His	Thr	del	Med+	EBV-1
cHL9	ND	Gly	Arg	Ser	/	/	/	del	Gly	Gly	His	Thr	del	Med+	EBV-1
cHL10	ND	Gly	Gln	Leu	WD	Gly	Gly	His	Gly	Gly	His	Thr	WD	B95-8	EBV-1
cHL11	NS	Gly	Pro	Leu	WD	Gly	Gly	His	Gly	Gly	His	Thr	WD	B95-8	EBV-1
cHL12	NS	Gly	Gln	Leu	WD	Gly	Gly	His	Gly	Gly	His	Thr	WD	B95-8	EBV-1
cHL13	TR	Gly	Gln	Leu	WD	Gly	Gly	His	Gly	Gly	His	Thr	WD	B95-8	EBV-1
cHL14	MC	Arg	Arg	Ser	/	/	/	del	Gly	Gly	His	Thr	del	Med+	EBV-1
cHL15	NS	Gly	Gln	Leu	WD	Gly	Gly	His	Gly	Gly	His	Thr	WD	B95-8	EBV-1
cHL16	NS	Gly	Gln	Leu	WD	Gly	Gly	His	Gly	Gly	His	Thr	WD	B95-8	EBV-1
cHL17	NS	Gly	Arg	Leu	/	Gly	Gly	del	/	/	His	Thr	del	Med+	EBV-1
cHL18	NS	Gly	Gln	Leu	WD	Gly	Gly	His	Gly	Gly	His	Thr	WD	B95-8	EBV-1
cHL19	MC	Gly	Gln	Leu	WD	Gly	Gly	His	Gly	Gly	His	Thr	WD	B95-8	EBV-1
cHL21	MC	Gly	Arg	Ser	/	/	/	del	Gly	Gly	His	Thr	del	Med+	EBV-1
cHL22	MC	Gln	Gln	Pro	WD	Gly	Gly	Asn	Gly	Gly	Pro	Thr	WD	NC	EBV-1
cHL24	MC	Gly	Arg	Ser	/	/	/	del	Gly	Gly	His	Thr	del	Med+	EBV-1
cHL25	MC	Gly	Pro	Leu	WD	Gly	Gly	His	Gly	Gly	His	Thr	WD	B95-8	EBV-1
cHL26	NS	Gly	Arg	Ser	WD	Gly	Gly	Arg	Gly	Gly	His	Thr	WD	Med−	EBV-1
cHL27	TR	Gly	Arg	Ser	/	/	/	del	Gly	Gly	His	Thr	del	Med+	EBV-1
cHL28	ND	Gly	Gln	Leu	WD	Gly	Gly	His	Gly	Gly	His	Thr	WD	B95-8	EBV-1
cHL29	NS	Gly	Gln	Leu	WD	Gly	Gly	His	Gly	Gly	His	Thr	WD	B95-8	EBV-1
cHL30	ND	Gly	Gln	Leu	WD	Gly	Gly	His	Gly	Gly	His	Thr	WD	B95-8	EBV-1
cHL31	NS	Gly	Gln	Leu	WD	Gly	Gly	His	Gly	Gly	His	Thr	WD	B95-8	EBV-1
cHL32	MC	Gly	Arg	Ser	/	/	/	del	Gly	Gly	His	Thr	del	Med+	EBV-1
cHL33	LD	Gly	Arg	Ser	WD	Gly	Gly	Arg	Gly	Gly	His	Thr	WD	Med−	EBV-1
cHL34	LR	Gly	Arg	Ser	/	/	/	del	Gly	Gly	His	Ala	del	Med+	EBV-1
cHL35	MC	Gly	Pro	Ser	/	/	/	del	Gly	Gly	His	Thr	del	Med+	EBV-1
cHL36	MC	Gly	Arg	Ser	/	/	/	del	Gly	Gly	His	Thr	del	Med+	EBV-1
cHL38	MC	Gly	Gln	Leu	WD	Gly	Gly	His	Gly	Gly	His	Ala	WD	B95-8/Med	EBV-1
cHL39	MC	Gly	Gln	Leu	WD	Gly	Gly	His	Gly	Gly	His	Thr	WD	B95-8	EBV-1
cHL40	ND	Gly	Gln	Leu	WD	Gly	Gly	His	Gly	Gly	His	Thr	WD	B95-8	EBV-1
cHL41	MC	Gly	Gln	Leu	WD	Gly	Gly	His	Gly	Gly	His	Thr	WD	B95-8	EBV-1
cHL42	MC	Gly	Arg	Leu	WD	Gly	Gly	His	Gly	Gly	His	Thr	WD	Unknown	EBV-1
cHL43	MC	Gly	Gln	Leu	WD	Gly	Gly	His	Gly	Gly	His	Thr	WD	B95-8	EBV-1
cHL44	LR	Gly	Gln	Leu	WD	Gly	Gly	His	Gly	Gly	His	Thr	WD	B95-8	EBV-1
cHL45	TR	Gly	Gln	Leu	WD	Gly	Gly	His	Gly	Gly	His	Thr	WD	B95-8	EBV-1
cHL46	ND	Gly	Gln	Leu	WD	Gly	Gly	His	Gly	Gly	His	Thr	WD	B95-8	EBV-1
cHL47	ND	Gly	Gln	Leu	WD	Gly	Gly	His	Gly	Gly	His	Thr	WD	B95-8	EBV-1
cHL48	ND	Gln	Gln	Pro	WD	Gly	Gly	Asn	Gly	Gly	His	Thr	WD	B95-8/NC	EBV-1
cHL49	MC	Gln	Gln	Pro	WD	Gly	Gly	Asn	Gly	Gly	His	Thr	WD	B95-8/NC	EBV-1
cHL50	ND	Gly	Arg	Ser	/	/	/	del	Gly	Gly	His	Thr	del	Med+	EBV-1
cHL52	LR	Gln	Gln	Pro	WD	Gly	Gly	Asn	Gly	Gly	Pro	Thr	WD	NC	EBV-1

LR—lymphocyte-rich, LD—lymphocyte-depleted, MC—mixed-cellularity, NS—nodular sclerosis, TR—SLL transformations to cHL, ND—subtype not determined; WD—without deletion, /—it is not possible to assess whether there is a characteristic change at this position specific to a certain subtype because it is part of 30 bp deletion that is characteristic for Med+, del—deletion, Med+—Mediterranean subtype with deletion, Med−—Mediterranean subtype without deletion, NC—North Carolina subtype; red-colored letters mark changes that result with amino acids at specific position in accordance with Edwards’ algorithm, while blue-colored letters mark changes at specific positions that result with amino-acids not included in Edwards’ algorithm [13].

**Table 3 ijms-23-15635-t003:** An algorithm for EBV LMP1 variant determination.

EBV Variants	Positions in the C-Terminus Region of *LMP1*											
	331	334	338	343–352	344	345	352	354	355	358	366	Deletion 30 bp
B95-8	Gly	Gln	Leu	without deletion	Gly	Gly	His	Gly	Gly	His	Ser	
China 1		Arg	Ser	deletion							Thr	
China 2	Gln		Pro		Asp				Ala/Thr		Thr	
Mediterranean		Arg	Ser				Arg/deletion				Asn/Thr/Ala	deletion/without deletion
Alaskan	Ala		Pro			Ser		Asp	Val			
North Carolina	Gln		Pro				Asn			Pro		

EBV, Epstein–Barr virus; LMP1, latent membrane protein-1.

## Data Availability

Available on request.

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
