# Peer review of "Molecular Characterisation of Epstein–Barr Virus in Classical Hodgkin Lymphoma"

_ijms, 2022, doi:10.3390/ijms232415635_

Round 1

Reviewer 1 Report

The current piece of work characterizes LMP1 variants classical Hodkin Lymphoma tumors collected from a Southeastern Europe population. Using a Sanger sequencing approach, they determined the EBV genotypes and LMP1 mutants from ~50 samples. This work provides a wealth of information regarding the molecular characteristics of EBV+ cHL, however, I have a few major concerns with the organization of the paper and how the data is interpreted. As it is, I would suggest some major re-writes and revisions to the data. Below are some comments and suggestions I believe may improve the impact of the work.

Comments and Suggestions to the authors:

1. The authors mention sequencing of the EBNA proteins, and specifically Sanger sequencing of EBNA3C to type the EBV strains associated with the LMP1 variants. This data should be included somewhere in the main text if not in a supplement material. In addition, a supplement could include pathology findings for each patient and their treatment. There are blanket statements made for all of these details, such as "they were treated with dox, bleo, vin, etc. Some did not achieve remission, some are female vs male, survival rates did not differ. While these are likely accurate assessments of the data, it would be better to include a comprehensive outline for readers to assess themselves and possibly relate back to their own work.

2. In Table 2 - The blue and red represent expected and unexpected changes in the LMP1 sequence. There is no discussion on what is "expected" or "unexpected". What is this referencing? Is it based off other findings (if so, the authors should address this somewhere)? 

3. In the abstract, the authors mention this study informing therapeutic development; however, there is no discussion on the relationship between their LMP1 variants and status of disease, prognosis factors, etc. I think to make that connection, there needs to be discussion of the molecular characteristics they found and how this could relate back to drug development? 

4. The impact of this work could be greater if, as mentioned above, the authors speak more to how these molecular changes in LMP1 relate to disease progression, prognosis, etc. There are many studies that have looked into these changes as they are happening at the C-terminus, some within the highly transformative CTAR2 domain is. In addition to their expected vs. unexpected analysis, it would be more interesting to also determine which variants result in a gain of function (over activate the NF-kB pathway) or loss of function (can no longer dimerize) etc. If not the molecular implications, at least some the of the clinical implications should be include. Some of this has been determined and could be referenced in the table and should be discussed. The method for the overall survival is described but the data is absent.

5. Was the whole LMP1 gene sequenced? Based on the methods, the EBNA3C amplicons were indicated, however, the LMP1 variants appears to be only for the C-terminus. It is not clear in the methods if the whole LMP1 gene was sequenced. It should be to get a full idea of the variants unless the authors want to justify why only the C-terminus was evaluated (again this could relate back to the biological function of the CTAR2 domain and transformation, but the authors have not made this clear).  **Primers used to amplify should be included in a table somewhere. 

6. Was there any correlation between the 5 patients that did not achieve remission and the molecular characteristics? It is mentioned but not expanded on in any way. 

Discussion - I would advise the authors to reorganize their discussion to communicate their results (and others) better and improve the impact of this work. Right now, it is one very long paragraph which references many studies and their outcomes, compared to their own. It took several reads and patience to get through. Another figure/table may be more appropriate given the amount of sample group sizes and statistics that are used. Alternatively, the authors could consider organizing the discussion to paragraphs dedicated to one observation. There is a lot of discussion of evidence for one thing followed by contrary findings. It is hard to find one cohesive story.

Author Response

Reviewer 1

The current piece of work characterizes LMP1 variants classical Hodkin Lymphoma tumors collected from a Southeastern Europe population. Using a Sanger sequencing approach, they determined the EBV genotypes and LMP1 mutants from ~50 samples. This work provides a wealth of information regarding the molecular characteristics of EBV+ cHL, however, I have a few major concerns with the organization of the paper and how the data is interpreted. As it is, I would suggest some major re-writes and revisions to the data. Below are some comments and suggestions I believe may improve the impact of the work.“

- Thank you for your observations. We accepted all your suggestions and made changes in the manuscript accordingly. We attached two copies of the revised manuscript: one with the track changes, and the other without them so that the final reading could be easier.

Comments and Suggestions to the authors:

  1. The authors mention sequencing of the EBNA proteins, and specifically Sanger sequencing of EBNA3C to type the EBV strains associated with the LMP1 variants. This data should be included somewhere in the main text if not in a supplement material. In addition, a supplement could include pathology findings for each patient and their treatment. There are blanket statements made for all of these details, such as "they were treated with dox, bleo, vin, etc. Some did not achieve remission, some are female vs male, survival rates did not differ. While these are likely accurate assessments of the data, it would be better to include a comprehensive outline for readers to assess themselves and possibly relate back to their own work.

Author's response: Thank you for the suggestion. EBNA3C genotyping is explained in Methods section (Determination of EBV types 1/2 and LMP1 variant determination) and results are given in the section Results (2.2. EBV variants and clinical data). The section about how was this determination done based on sequences gained by Sanger sequencing is rewritten, and additionally EBV-genotype and histological subtype data were added to table 2 according to Reviewer 2 suggestion.

We also added supplement Table S1. with available clinical and pathological data, and Supplement Table S2. with primer sequences used for amplification of EBNA3C and LMP1 amplicons.

  1. In Table 2 - The blue and red represent expected and unexpected changes in the LMP1 sequence. There is no discussion on what is "expected" or "unexpected". What is this referencing? Is it based off other findings (if so, the authors should address this somewhere)?

Author's response: We agree that "expected" and "unexpected" changes is vague description. We wanted to emphasise that in few cases our results could not follow Edwards’ algorithm and that changes in DNA sequence that we observed resulted in amino-acids not included in Edwards’ algorithm for those positions. We changed the explanation at the foot of the Table 2 and hope that now it is more clearly written.

  1. In the abstract, the authors mention this study informing therapeutic development; however, there is no discussion on the relationship between their LMP1 variants and status of disease, prognosis factors, etc. I think to make that connection, there needs to be discussion of the molecular characteristics they found and how this could relate back to drug development?

Author's response: Thank you for your observation. In order to consider our results in the context of drug development and novel LMP-targeted therapeutic strategies, in section Discussion, after the part on the distribution of EBV genotypes and LMP1 variants in cHL and other diseases, we added the following text and four appropriate references:

“Analysis of LMP1 molecular diversity in various malignant and non-malignant diseases associated with EBV is also important for the development of LMP1-targeted therapies including. Therapeutic strategies in this field include: (1) LMP1-specific cytotoxic T-cells as a part of adoptive cell transfer immunotherapy of EBV latency II-associated malignancies, (2) antibody-based immunotherapy using LMP1 peptide-specific immunisation in LMP1-expressing tumors, (3) immune checkpoint inhibitor therapy (blocking the biological function of an LMP1 target PD-L1 with monoclonal antibodies in recurrent or metastatic NPC), (4) LMP1 sequence-specific RNA-cleaving DNAzime (such as DZ509) and (5) development of small molecule inhibitors targeting LMP1 (such as B1.12, ACPLDLRSPCG peptide or nanoparticle-conjugated EBNA1-LMP1-binding peptide) [Ammous-Bou et al, 2019; Zha et al, 2020; Lo et al, 2021]. Of note, development of therapeutic vaccines involving either dendritic cell or viral vector-based strategies (or both) so far mainly targeted EBNA1 and LMP2 proteins (not LMP1) due to their immunogenicity [Jean-Pierre et al, 2021]. The results of our study have shown the presence of Mediterranean subtype with 30 pb deletion in approximately 28% of patients with cHL suggesting the need to evaluate the efficacy of the novel and/or developing LMP1-targeted therapies agains this high-risk LMP1 variant both during drug development stages and in clinical trials.”

Additional references:

Ammous-Boukhris, N.; Mosbah, A.; Ayadi, W.; Sahli, E.; Chevance, S.; Bondon, A.; Gargouri, A.; Baudy-Floc'h, M.; Mokdad-Gargouri, R. B1.12: a novel peptide interacting with the extracellular loop of the EBV oncoprotein LMP1. Sci Rep 2019, 9, 4389.

Zha, S.; Chau, H.F.; Chau, W.Y.; Chan, L.S.; Lin, J.; Lo, K.W.; Cho, W.C.; Yip, Y.L.; Tsao, S.W.; Farrell, P.J.; Feng, L.; Di, J.M.; Law, G.L.; Lung, H.L.; Wong, K.L. Dual-Targeting Peptide-Guided Approach for Precision Delivery and Cancer Monitoring by Using a Safe Upconversion Nanoplatform. Adv Sci (Weinh) 2021, 8, e2002919.

Lo, A.K.; Dawson, C.W.; Lung, H.L.; Wong, K.L.; Young, L.S. The Role of EBV-Encoded LMP1 in the NPC Tumor Microenvironment: From Function to Therapy. Front Oncol 2021, 11, 640207.

Jean-Pierre, V.; Lupo, J.; Buisson, M.; Morand, P.; Germi, R. Main Targets of Interest for the Development of a Prophylactic or Therapeutic Epstein-Barr Virus Vaccine. Front Microbiol,  2021, 12, 701611.

  1. The impact of this work could be greater if, as mentioned above, the authors speak more to how these molecular changes in LMP1 relate to disease progression, prognosis, etc. There are many studies that have looked into these changes as they are happening at the C-terminus, some within the highly transformative CTAR2 domain is. In addition to their expected vs. unexpected analysis, it would be more interesting to also determine which variants result in a gain of function (over activate the NF-kB pathway) or loss of function (can no longer dimerize) etc. If not the molecular implications, at least some the of the clinical implications should be include. Some of this has been determined and could be referenced in the table and should be discussed. The method for the overall survival is described but the data is absent.

Author’s responses: In response to the abovementioned comments, we added information on the changes in the transforming ability and immunogenicity associated with mutations in the C’ carboxy terminal region of LMP1 to the section Discussion and added appropriate references. In addition, we included the information on the prognostic value of LMP1 detection in HL in children and adults. Literature data on the impact of LMP1 variants on clinically-validated parameters including overall survival, risk of treatment failure or time of event-free survival are currently not available. Literature data on the high frequency of LMP1 variants with 30 bp deletion that exhibit highly transformative potential in vitro and in animal models, are already included in the section discussion. Due to extensive changes in section Discussion (based on your recommendations but also on recommendations from reviewer 1), please refer to the track changes in the revised version of the manuscript for specific details.   

 Also, follow up of patients in months is added to the Table S1. This was a basis for the comparison of survival rate between cHL patients with different EBV-1 subtypes.

  1. Was the whole LMP1 gene sequenced? Based on the methods, the EBNA3C amplicons were indicated, however, the LMP1 variants appears to be only for the C-terminus. It is not clear in the methods if the whole LMP1 gene was sequenced. It should be to get a full idea of the variants unless the authors want to justify why only the C-terminus was evaluated (again this could relate back to the biological function of the CTAR2 domain and transformation, but the authors have not made this clear). **Primers used to amplify should be included in a table somewhere.

Author's response: LMP1 variant determination is based on the analysis of the C'terminal part of the LMP1 gene, as recommended by Edwards, R.H; Seillier-Moiseiwitsch, F.; Raab-Traub, N. Signature amino acid changes in latent membrane protein 1 distinguish Epstein-Barr virus strains. Virology 1999, 261, 79-95. Since the analysis of the C-terminus is a universally accepted standard for the LMP1 variant analysis in our specific field, we used the abovementioned analytical approach. In section Supplement, we included Table S4 (based on suggestions on the revision of section Discussion) that presents a list of studies using the experimental approach identical to ours (focusing on the C’ terminus). In addition, analysis of the C’ terminal region of LMP1 is relevant for both molecular epidemiology of EBV LMP1 variants and, at the same time, allows the identification of a high-risk 30 bp deletion variant that is associated with increased oncogenic potential (based on in vitro data and data in animal models) and is epidemiologically associated with malignant diseases in specific models and geographic regions.

Based on your suggestion, we included a table with primers used in this study.

  1. Was there any correlation between the 5 patients that did not achieve remission and the molecular characteristics? It is mentioned but not expanded on in any way.

Thank you for your observation. This part was not clearly written. All patients that had available data about remission after the received therapy achieved full remission (23). We hope this is now more clearly written.

Additionally, 1 patient did not receive any therapy (per patient’s request), and one started the therapy but did not finish full treatment because of the complications. These patients are not included in final analysis about therapy outcome.

5 patients received therapy in our hospital, but were afterwards treated elsewhere and we have no data about the outcome of their treatment – for those 5 patients we were not able to evaluate remission and therefore we marked them in this regard as n. a. (not available).

Reviewer 2 Report

In this review manuscript, Begic and colleagues have identified EBV infection in a Hodgkin lymphoma-associated population in Southeast Europe, and further, the virus DNA sequence data obtained from those EBV-infected samples were utilized in the classification of EBV genotypes and their distribution. They compared their newly produced data with the previously reported EBV epidemiological studies.

This study makes a significant contribution to understanding EBV diversity in different populations and geographic locations, identifying viral, host, and environmental components that contribute to pathogenesis. The authors did an excellent job in the discussion section by doing a comprehensive analysis of the literature and bringing in epidemiological data from various geographical regions.

However, this manuscript might be improved further by addressing the concerns indicated below, which would benefit a broader audience of scientific readers.

 Critiques:

1.

This manuscript includes parts such as an Introduction, Results, and Methods, in addition to the thorough Discussion. Therefore, rather than a review article, the text is now presented as a research study article. I would recommend considering rewriting the Discussion section focused on their newly obtained data (adding a table to condense a lot of discussion text) and re-submitting it to the journal as a research article section.

2.

To ease the assimilation of the knowledge provided in the discussion section, the information contained in the discussion section should be presented in tabular format in addition to the text provided here.

 3.

The histological subtype data shown in Table 1 as groups might be added in Table 2 as an extra column, allowing the reader to connect the histological subtypes to individual EBV sequence/ genotype data.

Author Response

Reviewer 2.

Comment 1. This manuscript includes parts such as an Introduction, Results, and Methods, in addition to the thorough Discussion. Therefore, rather than a review article, the text is now presented as a research study article. I would recommend considering rewriting the Discussion section focused on their newly obtained data (adding a table to condense a lot of discussion text) and re-submitting it to the journal as a research article section.

and

Comment 2. To ease the assimilation of the knowledge provided in the discussion section, the information contained in the discussion section should be presented in tabular format in addition to the text provided here.

Author's response to comments 1 and 2 (partially overlapping):

- Thank you for pointing out mistake made during the submission process. The manuscript is indeed a research article. We will ask the assistant editor to make neccessary changes in the system, correct our mistake and fit our manuscript into the research article categories. 

- Based on your recommendation, Tables S3 and S4 have been added to the manuscript. Table S3 provides a summary of studies on the distribution of EBV types 1 and 2. Table 4 provides a summary of studies on the distribution of LMP1 variants. Both tables are appropriately referenced within the text. Since the information provided in the discussion has been also made available in the new Tables S3 and S4, we summarised a part of the text in a discussion to avoid the unnecessary duplicate information. Since the changes are substantial, we ask you to reffer to the corrected version of the manuscript with track changes for detailes. In brief, we pointed out to Tables S3 and S4 where appropriate, deleted significant parts of the text that could be covered by the information in the tables, except for studies specifically describing results on our research model (HL) and other malignant diseases (used for comparison). We hope that the new version of discussion and Tables S3 and S4 are helpful. Please note that parts of the discussion have been added (per request from the Reviewer 1).

Comment 3. The histological subtype data shown in Table 1 as groups might be added in Table 2 as an extra column, allowing the reader to connect the histological subtypes to individual EBV sequence/ genotype data.

Author's response to comment 3:

Information on the histological subtypes were added to Table 2, as well as EBV genotype determined according to EBNA3C (although it is the same for all cases - type 1, we added this column to emphasise this information according to the request by Reviewer 1). The new version of the Table 2 with track changes in provided.

Round 2

Reviewer 1 Report

The authors have addressed my previous concerns.